# Alcohol Consumption Reduces the Beneficial Influence of Protein Intake on Muscle Mass in Middle-Aged Korean Adults: A 12-Year Community-Based Prospective Cohort Study

**DOI:** 10.3390/nu11092143

**Published:** 2019-09-07

**Authors:** Eunjin So, Hyojee Joung

**Affiliations:** 1Department of Clinical Nutrition, The Catholic University of Korea Seoul St, Mary’s Hospital, Seoul 06591, Korea; 2Department of Public Health, Graduate School of Public Health& Institute of Health and Environment, Seoul National University, Gwanak-gu, Seoul 08826, Korea

**Keywords:** protein intake, muscle mass, alcohol consumption, middle-aged, Korean Genome and Epidemiology Study (KoGES), cohort study

## Abstract

The influence of alcohol consumption on the association of protein intake with muscle mass was assessed using data from the Korean Genome and Epidemiology Study. Dietary protein intakes of 4412 middle-aged participants with normal baseline muscle mass were assessed using a semi-quantitative Food Frequency Questionnaire, and baseline alcohol consumption data (e.g., frequency and amount) were collected using a structured questionnaire. The skeletal muscle mass index (SMI), defined as the weight-adjusted skeletal muscle mass, was measured using multi-frequency bioelectrical impedance analyses every 2 years until the study endpoint. Low muscle mass was defined as a SMI <2 standard deviations below the sex-specific normal mean for a young reference group. During a 12-year follow-up, 395 subjects developed a low SMI. After multivariate adjustments, high protein intake (≥1.2 g/kg body weight (BW)) was shown to reduce the risk of low SMI development in both men (hazard ratio (HR): 0.24; 95% confidence interval (CI): 0.12, 0.51; *p* for trend < 0.001) and women (HR: 0.29; 95% CI: 0.16, 0.53; *p* for trend < 0.001), compared with low protein intake (<0.8 g/kg BW). Alcohol consumption attenuated the protective influence of protein intake against low SMI development in women (HR: 0.64; 95% CI: 0.18, 2.25; *p* for trend = 0.478). Among the total subjects, heavy drinkers with high protein intake were not significantly associated with the development of a low SMI (HR: 0.20; 95% CI: 0.03, 1.50; *p* = 0.117). Additional research should clarify the dose-response effects of alcohol consumption on muscle mass relative to daily protein intake.

## 1. Introduction

The progressive loss of muscle mass is the primary factor used to determine sarcopenia, a syndrome characterized by low muscle mass and strength [1]. Sarcopenia has been associated with increased risks of falls and fractures, reduced cardiopulmonary function, metabolic syndrome, and insulin resistance, and eventually leads to the disability, hospitalization, and death of older individuals [2].

According to the World Health Organization, the number of individuals older than 60 years of age is expected to increase from 900 million to 2 billion between 2015 and 2050 [3]. In other words, the population of elderly individuals who will be exposed to the risks of deteriorating strength and mobility associated with muscle loss is expected to increase enormously. According to recent estimates, the prevalence of sarcopenia ranges from 5% to 13% among individuals aged 60–70 years and from 11% to 50% among those aged ≥80 years [4]. Age-related sarcopenia is a common condition associated with significant personal and financial burdens. One prospective study estimated that sarcopenia would increase hospitalization costs by 58.5% and 34% for patients <65 and ≥65 years of age, respectively [5].

The etiology of sarcopenia is multifactorial. Although aging is the leading cause of sarcopenia, this condition can be accelerated by modifiable lifestyle factors, such as physical inactivity, alcohol consumption, smoking, and undernutrition [6]. To date, protein intake, in terms of both the quantity and quality, has been the main nutritional focus [7]. However, the evidence increasingly suggests that the current recommended daily allowance (RDA) of protein, 0.8 g/kg body weight (BW), might not be adequate to maintain lean mass and prevent functional declines among the elderly [8,9]. Rather, recent reviews and consensus statements have suggested that a protein intake of 1.0–1.5 g/kg BW may confer health benefits beyond those afforded by simply meeting the minimum RDA [10,11,12].

The potentially important role of alcohol consumption in the prevention of age-related muscle loss has not been fully investigated. Some experimental animal studies have demonstrated a relationship between alcohol consumption and sarcopenia [13,14,15]. Specifically, alcohol consumption inhibited the synthesis of skeletal muscle proteins in rats. In humans, this relationship remains controversial, although a negative relationship has been reported between alcohol consumption and sarcopenia in the general population [16,17,18]. Moreover, little is known about the potential interacting effects of alcohol consumption and protein intake on sarcopenia [19].

Therefore, this prospective cohort study investigated the influences of various recommended protein intake levels on the development of low muscle mass according to alcohol consumption in middle-aged individuals.

## 2. Materials and Methods

### 2.1. Study Population

The data included in this study were obtained from a large community-based cohort based on the Korean Genome and Epidemiology Study (KoGES) [20]. For KoGES, the participant eligibility criteria included an age of 40–69 years and residence in Ansan (urban) or Ansung (rural), Korea, for at least 6 months before enrolment. Baseline examinations were performed in 2001 and 2002, and follow-up examinations were repeated every 2 years through 2014. Of the original 10,030 participants, 4412 remained in the final analysis after excluding 2417 who did not complete the baseline Food Frequency Questionnaire (FFQ) or who had incomplete anthropometric data, 56 with abnormally low or high daily energy intakes (<500 or >5000 kcal), 3042 who did not participate in the follow-up examinations, and 103 with a low skeletal muscle mass index (SMI) at baseline. Written informed consent was obtained from all participants. All procedures were approved by the Institutional Review Board of the Catholic Medical Center (No. KC17ZESI0645).

### 2.2. Assessment of Dietary Intake

At baseline, the participants’ usual dietary intakes were assessed by trained dietitians using a validated 103-item semi-quantitative FFQ [21]. Nine response options were provided to describe the frequency of consumption of each food (never or almost never, 1 time/month, 2–3 times/month, 1–2 times/week, 2–3 times/week, 3–4 times/week, 5–6 times/week, 1 time/day, 2 times/day, or 3 times/day), and three response options were provided to describe the portion size (1/2 serving, 1 serving, and ≥2 servings). To enhance the accuracy of serving size recall, pictures of each food item were used as references. For each participant, the daily protein and other nutrient intakes were estimated based on the sum of the intake of each food item according to The Food Composition Database (Seoul, Korea: The Rural Development Administration, 2007).

### 2.3. Measurement of Alcohol Consumption 

Total alcohol intake was assessed by questioning the participant about his or her alcohol consumption habits during the month before the interview. Participants were asked about the frequency of consumption, amount per serving, and drinking glass size. Six response options were provided to describe the frequency of alcohol consumption (1 time/month, 2–3 times/month, 1 time/week, 2–3 times/week, 4–6 times/week, or >7 per week). The amounts of beer, wine, spirits, Korean distilled spirits, raw rice wine, and refined rice wine were recorded. Total alcohol intake was calculated as the sum of the values converted to the amount of pure alcohol consumed per day for all six beverages. The participants were divided into three groups depending on the amount of alcohol consumed per day: non-drinkers, light-to-moderate drinkers (1‒40 g/day for men, 1–20 g/day for women), and heavy drinkers (≥40 g/day for men, ≥20 g/day for women). Binge drinking was defined as the consumption of ≥5 alcoholic drinks for men or ≥4 alcoholic drinks for women on at least one occasion during the previous 30 days, and participants were subcategorized into two groups based on this variable: social drinkers (<1 time/month) and binge drinkers (≥1 time/month) [18].

### 2.4. Measurement of Body Composition

Fat mass and lean mass (i.e., fat-free mass) were assessed using multi-frequency bioelectrical impedance analysis (MF-BIA; InBody 3.0; Biospace, Seoul, Korea) according to standard procedures. The skeletal muscle mass was estimated by dividing the total lean mass by 0.52 [22]. In our study, a low muscle mass was defined using the SMI, which was calculated as the total skeletal muscle mass (kg)/weight (kg) × 100.This measure adjusts for stature and the mass of non-skeletal muscle tissues (fat, organs, and bone), as described by Janssen et al. [23]. We used SMI cutoff points of 35.71% for men and 30.70% for women to define low muscle mass, based on a threshold of <2 standard deviations below the sex-specific normal means for a young reference group as defined in previous Korean studies [24]. The same method was used to determine the incidence of low muscle mass every 2 years until the study endpoint. The permitted range for each follow-up measurement was 24 ± 12 months, while the actual average range was 23.4 ± 0.8 months.

### 2.5. Covariates

The participants’ demographic characteristics and medical histories were obtained using an interviewer-administered questionnaire. The questionnaire included items concerning sex, age, marital status, education, smoking habits, regular physical activity, self-perceived dental health status, chronic diseases, and residential area at baseline. To evaluate the influence of protein intake on age-related muscle loss in younger versus older adults, age was dichotomized as younger or older than 60 years. Marital status was categorized as married and unmarried (including divorced, separated, and others). Education level was categorized as high school or lower and college or higher. The smoking status was used to classify participants as smokers (current smokers) and non-smokers (former smokers and non-smokers). Regular physical activity was recorded as ”yes“ if the participant performed ≥2.5 h of exercise per week according to the global World Health Organization recommendation [25,26]. The self-perceived dental health status was categorized as poor or other (good and fair). The presence or absence of chronic diseases, such as myocardial infarction, congestive heart failure, coronary artery disease, peripheral arterial disease, cerebrovascular disease, asthma, chronic obstructive pulmonary disease, cancer, dementia, and arthritis was recorded.

### 2.6. Statistical Analysis

The weight-adjusted protein intake was categorized according to three different nutritional recommendations: RDA (≤0.8 g/kg BW) [27], International Study Group, to review dietary protein needs with aging (PROT-AGE Study Group recommendation) (0.81‒1.19 g/kg BW) [10], and Nordic Nutrition Recommendation 2012 (≥1.2 g/kg BW) [28]. For this study, these three recommendations were used to define the low, moderate, and high protein intake groups, respectively, and the low intake group was used as the reference. The baseline characteristics of the study participants are reported as percentages, means, and standard deviations and were compared using the Mantel-Haenszel χ^2^ test for categorical variables and linear regression analyses for continuous variables. A one-way analysis of variance was used to test any intergroup differences in the baseline measures and percentage changes. A Cox proportional hazards regression model was used to calculate the hazard ratios (HRs) and 95% confidence intervals (CIs) of total protein intake associated with the development of a low SMI during follow-up according to alcohol consumption. Age, skeletal muscle mass, energy intake, marital status, education, income, smoking, regular physical activity, self-perceived dental health status, chronic disease, and residential area at baseline were included as covariates in this model. To test for linear trends across the protein intake categories, we created a continuous variable using the median total protein score in each group. IBM SPSS Statistics for Windows, version 24.0 (IBM Corp., Armonk, NY, USA) was used for all statistical analyses. A two-sided *p*-value of <0.05 was considered statistically significant.

## 3. Results

During a median follow-up of 141 months (range: 19–152 months), we observed an incidence of newly developed low lean mass of 9.5% (395 cases). Table 1 presents the baseline characteristics of the study participants according to the total protein intake at baseline. Overall, 26.8% of the participants had low protein intake (≤0.8 g/kg BW), 44.2% had a moderate intake (0.81–1.19 g/kg BW), and 28.9% had high protein intake (≥1.2 g/kg BW). Both men and women in the high protein intake group had a higher educational level (*p* = 0.005 and <0.001, respectively), earned a higher income (*p* = 0.024 and <0.001, respectively), and were physically active (*p* < 0.001 and 0.004, respectively). Men with high protein intakes were more likely to be current drinkers (*p* = 0.074), heavy drinkers (*p* = 0.019), and binge drinkers (*p* = 0.033) and to consume alcohol more frequently (*p* = 0.001). Women with high protein intakes were significantly younger (*p* < 0.001). The smoking status was not significantly associated with protein intake. Men and women with high protein intake reported higher intakes of energy (*p* < 0.001) and energy from fat (*p* < 0.001) and a lower intake of energy from carbohydrates (*p* < 0.001), compared to those with a low protein intake. Regarding body composition, both men and women with high protein intake had a lower weight (*p* < 0.001), body mass index (BMI) (*p* < 0.001), fat mass (*p* < 0.001), and lean mass (*p* < 0.001) at baseline than those with a low protein intake. However, high protein intake was associated with a high SMI (*p* < 0.001) in men but a low SMI in women (*p* < 0.001).

Figure 1 shows the percent changes in body composition over the 12-year follow-up period. Both men and women with higher protein intake exhibited lower reductions in weight (p = 0.003 and < 0.001, respectively) and lean mass (p = 0.001 and < 0.001, respectively) during follow-up. By contrast, the mean changes in fat mass (p = 0.079 for men, 0.009 for women) and BMI (p = 0.002 for men, < 0.001 for women) increased significantly in men and women with a high protein intake.

Table 2 presents the hazard ratios (HRs) and 95% confidence intervals (95% CIs) of the risk of developing a low SMI for each baseline dietary protein intake level, according to alcohol consumption. After adjusting for covariates, a high total protein intake (≥1.2 g/kg BW) decreased the risk of developing a low SMI in both men (HR: 0.24; 95% CI: 0.12, 0.51; p for trend < 0.001) and women (HR: 0.29; 95% CI: 0.16, 0.53; p for trend < 0.001) relative to a low protein intake. Women with a moderate total protein intake (0.81–1.19 g/kg BW) had a decreased risk of developing a low SMI relative to those with low protein intake after adjusting for covariates (HR: 0.54; 95% CI: 0.38, 0.76; p for trend < 0.001). In covariate-adjusted subgroup analyses, high protein intake was associated with a lower risk of developing a low SMI relative to low protein intake in both men (HR: 0.28; 95% CI: 0.07, 1.09; p for trend = 0.064) and women (HR: 0.23; 95% CI: 0.11, 0.45; p for trend < 0.001) who did not consume alcohol. However, these associations disappeared in the subgroup of women who consumed alcohol (HR: 0.64; 95% CI: 0.18, 2.25; p for trend = 0.478) after adjusting for covariates. Among the total subjects, the associations between protein intake and the development of low muscle mass were significant at ≥0.81 g/kg BW for non-drinkers (HR: 0.55; 95% CI: 0.39, 0.77; p = 0.001), those who consumed alcohol <1 time/week (HR: 0.60; 95% CI: 0.44, 0.81; p = 0.001), and social drinkers (HR: 0.59; 95% CI: 0.44, 0.78; p < 0.001). However, among the total subjects, heavy drinkers with high protein intake were not significantly associated with the development of a low SMI (HR: 0.20; 95% CI: 0.03, 1.50; p = 0.117).

## 4. Discussion

To our knowledge, this was the first large community-based prospective cohort study to investigate the association between protein intake and muscle mass according to alcohol consumption in Korean adults. We found that both men and women with a protein intake exceeding 1.2 g/kg BW had a lower risk of developing low SMI, compared to those with a protein intake of <0.8 g/kg BW. However, the beneficial influences of high protein intake on the prevention of a low SMI were not observed in women who consumed alcohol. Among the total subjects, heavy drinkers with high protein intake (≥1.2 g/kg BW) were not significantly associated with the development of a low SMI.

Consistent with previous human studies, men and women with higher protein intake had significantly lower decreases in lean mass [29,30]. In an older person, weight loss would be expected to cause losses in both lean mass and fat mass [31,32]. In the 12-year follow-up period of this study, the weight and lean mass decreased, while the fat mass and BMI increased. Recently, however, research has demonstrated that high protein intake could effectively increase the lean mass while maintaining or decreasing the body weight and fat mass. A meta-analysis of randomized controlled trials showed that older men and women who consumed a higher protein diet (vs. a normal protein level) exhibited better lean mass retention even while losing body mass during periods of diet-induced energy restriction [33]. In another meta-analysis of randomized controlled trials, Wycherley et al. reported that when compared with an energy-restricted standard protein diet, an isocaloric high protein diet provided modest benefits in terms of reductions in the body weight, fat mass, and triglycerides and mitigated the reductions in lean mass and resting energy expenditure [34]. Further community-based large cohort studies or trials are needed to clarify how protein intake affects overall body composition, including fat mass and weight.

In this study, higher protein intake was associated with a decreased risk of developing a low SMI in both men and women, after adjusting for covariates and the SMI at baseline. This finding was consistent with those of previous large prospective studies in which the dietary protein intake was identified as the main nutritional factor associated with the preservation of muscle mass and maintenance of physical function [29,30,31,32]. For example, Houston et al. reported that among community-dwelling older adults in the highest protein intake quintile (1.1 g/kg BW), the total losses in lean body mass and appendicular lean mass over 3 years were approximately 40% lower than those of adults in the lowest quintile (0.7 g/kg BW) [29]. Another 5-year cohort study found that participants in the highest protein intake tertile (>87 g/day) had whole-body and appendicular lean mass values 5.4–6.0% higher than those of participants in the lowest tertile [30]. Moreover, in a meta-analysis of adult men and women with a mean age of ≥50 years, older adults who consumed a higher protein diet (≥1.0 g/kg/day) preserved more lean mass during intentional weight loss, compared to those who consumed a lower protein diet (<1.0 g/kg/day) [33]. The current study demonstrated significant associations at protein intake levels of ≥0.81 g/kg BW for women and ≥1.2 g/kg BW for men. These findings seem to support the hypothesis that the current RDA of protein (0.8 g/kg BW) is insufficient to promote optimal health and preserve physical performance in older men [8,9]. Increasing evidence suggests that a protein intake of 1.0–1.2 g/kg BW may benefit older adults by preventing or mitigating sarcopenia [10,11,12]. Further, Suominen et al. suggested that older individuals suffering from illness, physiological stress, or sarcopenia have higher requirements for protein intake (1.2–1.5 g/kg BW) than their healthy counterparts (1–1.2 g/kg BW) [34]. Moreover, according to data from the 2008–2012 Korean National Health and Nutrition Examination Survey (KNHANES), 18.8% of adults aged ≥60 years and 34.9% of adults aged ≥70 years consumed less than the estimated average requirements (40 g/day for men and 35 g/d for women); in other words, many older Korean adults consume insufficient amounts of protein [35]. Additional evidence from high-quality, multicenter clinical trials is needed to assess the long-term influences of increased protein intake on the preservation of muscle health in older adults.

In the present study, we observed a protective influence of high protein intake against the development of a low SMI in both men and women who did not consume alcohol, whereas this influence disappeared in women who consumed alcohol. Previous studies have attempted to explain the molecular mechanism underlying alcohol-induced muscle damage [36,37,38]. Excessive exposure to alcohol induces defects in muscle tissue, including the formation of aldehyde protein adducts, losses of ribosomes and myofibrillary proteins, a reduction in protein synthesis, and an increase in the level of sarcoplasmic-endoplasmic reticulum Ca2+-ATPase [36]. The habitual consumption of extreme amounts of alcohol (>80 g alcohol/day) can lead to chronic alcoholic myopathy and the development of muscle weakness and wasting [38]. In animal studies, ethanol impaired skeletal muscle protein synthesis and increased muscle autophagy [13,15,39]. Unfortunately, the few studies that have examined the association between alcohol and muscle mass in humans have not yielded consistent findings [17,18]. A recent meta-analysis of a population of non-cancer patients did not support alcohol consumption as a risk factor for sarcopenia [16]. However, that meta-analysis included studies that had been designed to consider the relationship between alcohol consumption and muscle mass as the primary endpoint. In contrast, our study was designed to focus on the interaction of alcohol consumption with the relationship between SMI and protein intake. In the current study, high protein intake did not have a beneficial influence on the prevention of a low SMI among women who consumed alcohol. This finding is consistent with a recent Korean study of postmenopausal women in which participants in the high-risk alcohol consumption group, defined as those with Alcohol Use Disorders Identification Test scores ≥15, had a higher risk of sarcopenia [17]. However, another Korean study of men aged 60 years did not observe a difference in the proportion of participants who consumed alcohol between the groups with and without sarcopenia [40]. Another study of older men in France did not identify an association between alcohol intake and sarcopenia [41]. Furthermore, a recent cross-sectional Korean study found that alcohol consumption was associated with sarcopenia in women, but not in men [18]. The authors of that study explained that women experience higher blood alcohol concentrations because their bodies contain smaller volumes of water in which to dilute the alcohol. The bodies of elderly women contain even smaller volumes of water, leading to a decreased tolerance for alcohol and a slower rate of alcohol metabolism [42]. Therefore, our study and previous studies suggest that sex may affect the association between alcohol consumption and muscle mass. Although we could not identify the influences of alcohol consumption on the relationship between protein intake and the development of a low SMI in men, the protective influence of high protein intake against the development of a low SMI disappeared among heavy drinkers in total subjects adjusted for sex. Further studies are needed to examine these sex-specific associations and explore the mechanisms of underlying sex differences in the relationship between alcohol consumption and SMI.

The present study had several strengths, including a large cohort and a long follow-up period, which enabled an investigation of the associations of habitual dietary protein intake and alcohol consumption with the SMI in Korean adults. However, our study also had a few limitations. First, bioelectrical impedance analysis (BIA), a non-invasive method used to assess skeletal muscle mass, is useful in large population-based studies; however, the results may be affected by several factors, including age, hydration status, food or beverage consumption, and exercise intensity. To reduce the possibility of measurement errors, the participants fasted overnight prior to the BIA, and the researchers confirmed any recent history of intensive exercise, bathing, or excessive sweating. In addition, BIA has been validated for the estimation of body composition, using dual energy X-ray absorptiometry (DXA) as a reference standard [43,44,45]. The European Working Group on Sarcopenia in Older People (EWGSOP) also suggested that the BIA is a portable alternative to DXA [1]. Second, we assessed the dietary protein intake and alcohol consumption only at baseline and did not determine whether these variables changed over time. Third, alcohol consumption by women is relatively poorly accepted in Korean culture; accordingly, female participants may have under-reported their alcohol consumption and drinking pattern scores. Fourth, the combination of alcohol abstainers and former alcohol drinkers into a current non-drinkers group might have introduced bias and masked the real effects of alcohol abstention. Finally, the small sample of heavy drinkers might have led to sampling bias and affected the statistical significance. To confirm this result, future large community-based cohort studies or trials should address alcohol consumption and its relationship with muscle mass.

## 5. Conclusions

In summary, we observed that men and women with protein intake levels exceeding 1.2 g/kg BW had a lower risk of developing a low SMI, compared to those with protein intake levels <0.8 g/kg BW. However, alcohol consumption attenuated the beneficial influences of high protein intake against the development of a low SMI in women. In addition, among the total subjects, the association between protein intake and SMI was not detected in heavy drinkers. Our findings suggest that alcohol consumption might reduce the muscle-preserving influences of dietary protein. More comprehensive studies are needed to clarify the dose-response effects of alcohol consumption on the relationship between daily protein intake and declining muscle mass. 

## Figures and Tables

**Figure 1 nutrients-11-02143-f001:**
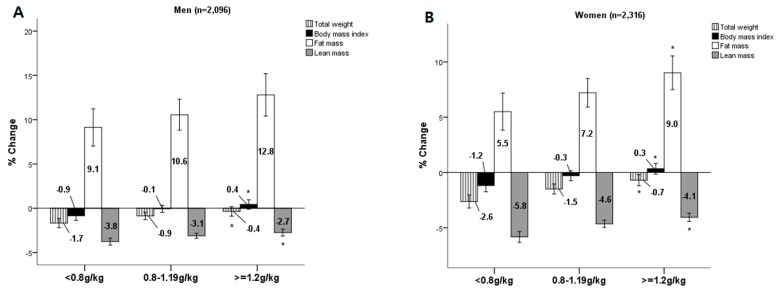
Change in body composition over 12 years by the levels of dietary protein intake at baselines. (**A**) men and (**B**) women. % Change = ((12-year follow-up value-baseline value)/baseline value × 100). ANOVA = analysis of variance. * Indicates significantly linear trend across the protein intake categories according to different recommendations: *p* for trend < 0.05.

**Table 1 nutrients-11-02143-t001:** General characteristics of study participants by the levels of total protein intake at baseline.

Protein Intake	Men (*n* = 2096)				Women (*n* = 2316)			
	≤0.8 g/kg BW	0.81‒1.19 g/kg BW	≥1.2 g/kg BW	*p* for Trend ^1^	≤0.8 g/kg BW	0.81‒1.19 g/kg BW	≥1.2 g/kg BW	*p* for Trend
	*n* = 538	*n* = 980	*n* = 533		*n* = 601	*n* = 972	*n* = 743	
Demographics and lifestyles (%)								
Age (years, mean ± SD)	50.5 ± 8.0	49.7 ± 7.7	50.5 ± 8.2	0.073	52.4 ± 8.6	51.1 ± 8.4	49.6 ± 8.2	0.000
Residential area (city)	61.6	74.4	64.0	0.301	51.4	65.5	60.3	0.003
Educational (≥College)	19.9	27.8	27.1	0.005	4.0	7.5	11.5	0.000
Household income (≥3,000,000 KRW)	21.7	31.0	27.5	0.024	11.2	16.2	22.5	0.000
Marital status (married)	96.9	96.8	97.4	0.673	85.2	87.5	90.7	0.002
Smoking (yes)	45.5	42.3	47.8	0.475	2.9	2.0	2.3	0.547
Chronic disease (yes)	1.5	1.1	1.9	0.665	5.7	4.4	3.9	0.132
Dental health status (poor)	36.3	39.2	40.1	0.189	45.5	43.0	37.7	0.004
Regular physical activity (yes)	12.5	19.2	22.0	0.000	14.1	17.3	20.1	0.004
Alcohol drinking (yes) ^2^	71.6	73.7	76.3	0.074	24.7	25.9	28.2	0.142
Heavy drinking (yes)	13.7	14.6	18.1	0.019	1.5	1.4	1.2	0.336
Drinking frequency (≥1 time/week)	49.1	52.8	58.9	0.001	7.0	7.6	9.2	0.136
Binge drinker (yes) ^3^	42.2	49.4	48.4	0.033	6.7	7.1	5.7	0.421
Dietary intake (Mean ± SD)								
Energy (kcal/day)	1528.9 ± 271.5	1961.5 ± 312.4	2628.2 ± 574.4	0.000	1345.7 ± 291.4	1737.9 ± 296.0	2442.1 ± 617.0	0.000
Carbohydrate (% of energy)	74.0 ± 5.0	69.0 ± 5.3	65.0 ± 6.4	0.000	76.3 ± 5.3	71.9 ± 5.5	68.2 ± 6.7	0.000
Fat (% of energy)	12.4 ± 4.1	16.0 ± 4.1	19.0 ± 4.8	0.000	10.4 ± 4.2	13.7 ± 4.4	16.6 ± 5.0	0.000
Protein (% of energy)	12.3 ± 1.7	13.8 ± 1.8	15.3 ± 2.3	0.000	11.9 ± 1.9	13.3 ± 1.8	14.7 ± 2.2	0.000
Protein (g/day)	46.6 ± 9.2	67.3 ± 11.3	99.2 ± 22.7	0.000	39.4 ± 8.2	58.4 ± 9.4	88.6 ± 24.1	0.000
Protein (g/kg body weight)	0.7 ± 0.1	1.0 ± 0.1	1.5 ± 0.3	0.000	0.7 ± 0.1	1.0 ± 0.1	1.6 ± 0.4	0.000
Body composition (Mean ± SD)								
Weight (kg)	71.0 ± 9.5	68.4 ± 9.0	65.4 ± 8.7	0.000	61.4 ± 8.1	59.1 ± 7.4	56.2 ± 7.2	0.000
Body Mass Index (kg/m^2^)	25.1 ± 2.8	24.4 ± 2.7	23.5 ± 2.7	0.000	25.7 ± 3.0	24.8 ± 2.8	23.6 ± 2.7	0.000
Fat mass (kg)	16.1 ± 4.7	15.1 ± 4.5	13.6 ± 4.2	0.000	20.1 ± 4.9	18.8 ± 4.4	17.1 ± 4.3	0.000
Lean mass (kg)	54.80 ± 6.2	53.3 ± 6.1	51.8 ± 5.9	0.000	41.2 ± 4.3	40.3 ± 4.3	39.1 ± 4.1	0.000
Skeletal muscle mass index (%) ^4^	40.3 ± 2.3	40.7 ± 2.4	41.4 ± 2.5	0.000	35.2 ± 2.6	35.6 ± 2.4	26.4 ± 2.5	0.000

Abbreviations: BW, body weight; SD, standard deviations; KRW, Korean won. ^1^ P for trend was calculated from a linear regression analysis for continuous variables and Mantel-Haenszel χ^2^ for categorical variables; ^2^ “Yes” was defined as a person who answered, “No” to the question “Do you never drink alcohol or drink from the beginning?”; ^3^ Binge drinker was defined as a participant who drinks more than once a month of binge drinking; ^4^ Skeletal muscle mass index (%) = total skeletal muscle mass (kg)/weight (kg) × 100.

**Table 2 nutrients-11-02143-t002:** Hazard ratios (HRs) and 95% confidence intervals (95% CIs) for the risk of developing low SMI by the levels of dietary protein intake at baseline according to alcohol consumption ^1^.

Alcohol Consumption Status	<0.8 g/kg BW		0.8‒1.19 g/kg BW			≥1.2 g/kg BW			*p* for Trend
	Case(*n*)/Person-Months	Reference	Case(*n*)/Person-Months	HR	95% CI	Case(*n*)/Person-Months	HR	95% CI	
**Men (*n* = 2096)**	51/81,130	1.00	88/135,815	0.70	(0.47, 1.05)	32/74,611	0.24	(0.12, 0.51)	0.000
Non-drinkers	13/22,991	1.00	24/35,727	0.85	(0.39, 1.84)	10/17,684	0.28	(0.07, 1.09)	0.064
Drinkers	43/57,861	1.00	64/99,950	0.66	(0.41, 1.07)	22/56,784	0.23	(0.10, 0.54)	0.001
**Women (*n* = 2316)**	89/82,518	1.00	87/135,645	0.54	(0.38, 0.76)	42/104,027	0.29	(0.16, 0.53)	0.000
Non-drinkers	74/61,493	1.00	67/99,664	0.48	(0.32, 0.70)	31/74,485	0.23	(0.11, 0.45)	0.000
Drinkers	15/20,737	1.00	19/35,141	0.90	(0.40, 2.03)	11/29,260	0.64	(0.18, 2.25)	0.478
**Total (*n* = 4412)** **Quantity of Drinking**									
Non-drinkers	87/84,484	1.00	91/135,391	0.55	(0.39, 0.77)	41/92,169	0.24	(0.13, 0.44)	0.000
Light-to-moderate drinkers	49/64,233	1.00	65/109,753	0.68	(0.44, 1.07)	25/68,138	0.27	(0.12, 0.58)	0.158
Heavy drinkers	7/12,135	1.00	14/20,919	0.68	(0.22, 2.12)	6/14,312	0.20	(0.03, 1.50)	0.024
**Frequency of drinking**									
<1 time/week	112/117,898	1.00	124/189,519	0.60	(0.44, 0.81)	54/125,281	0.27	(0.16, 0.47)	0.000
≥1 time/week	34/45,750	1.00	51/81,941	0.71	(0.42, 1.20)	20/53,357	0.28	(0.11, 0.70)	0.007
**Presence of binge drinking**									
Social drinkers ^2^	119/123,801	1.00	128/194,703	0.59	(0.44, 0.78)	63/136,522	0.28	(0.17, 0.47)	0.000
Binge drinkers ^3^	27/39,847	1.00	47/76,757	0.77	(0.41, 1.43)	11/42,116	0.26	(0.08, 0.81)	0.018

Abbreviations: CI, confidence interval; HR, hazards ratio; KRW, Korean won. Estimated using the Cox proportional hazards regression model. ^1^ Adjusted for age (<60/≥60 years), skeletal muscle mass at baseline, energy intake, marital status (married/others), education (≥college/others), income (≥3,000,000 KRW per month/other), smoking (yes/no), regular physical activity (yes/no), self-perceived dental health status (poor/others), chronic disease (yes/no), residential area (urban/rural). ^2^ Social drinker was defined as a participant who drinks less than once a month of binge drinking. ^3^ Binge drinker was defined as a participant who drinks more than once a month of binge drinking.

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
