# Peer review of "Alcohol Consumption Reduces the Beneficial Influence of Protein Intake on Muscle Mass in Middle-Aged Korean Adults: A 12-Year Community-Based Prospective Cohort Study"

_nutrients, 2019, doi:10.3390/nu11092143_

Round 1
Reviewer 1 Report
This is an interesting paper that suggests that ethanol ingestion results in sarcopenia prospectively and to a greater extent in women than in men. The authors have used a large cohort of subjects in whom data was collected prospectively. Despite some concern about the use of BIA, these data are of potential interest.
A few concerns limit enthusiasm for the study.
Inbody is really a weak method to quantify body composition, and the authors did not use a segmental BIA. Also, phase angle has been suggested to be a robust indicator of muscle mass. However, this cannot be an argument since the authors used an existing data set but this may be considered in the Discussion section that more reliable measures of body composition were not used but the use of a similar methodology prospectively overcomes this limitation. However, they must clarify if the same method was used over the 12 year follow up period. The authors state that 2 yearly evaluations were performed but in human studies, it is difficult to have exactly the same intervals between studies. What was the permitted range for each measurement e.g. 2y ±3 m/6 m etc? Typographical errors need to be reviewed as should the grammar that is interspersed throughout the manuscript. E.g. line 256, high protein intake on prevention of (not development of low smi); line 304 observe or report (not observed) etc. P<0.05 was predefined significance level, so stating that p<0.074, line 159 is not significant. Lines 158-166 in the Results section is very confusing. Higher protein intake was associated with higher ethanol consumption- yet, higher protein intake was associated with better SMI, interpretation of this observation is not explained. Lower lean body mass and yet higher SMI, how is this explained, is LBM not the measure from BIA that is used to calculate the SMI? Line 311, there is a recent JBC paper that shows that ethanol is directly metabolized by the muscle and can explain how ethanol can result in muscle loss. They may consider citing this very recent work on ethanol and ammonia synergism on muscle protein homeostasis.Author Response
We thank you for your time and expertise in reviewing the manuscript.
All changes have been made according to the reviewer’s comments in the text. Detailed explanations we have made are described in the rebuttal letter we enclosed.
The English in this manuscript has been checked by at least two professional editors, both native speakers of English.
Thank you for your consideration for our manuscript.
Yours sincerely,

Reviewer 2 Report
Dear Author,
Thank you for your work. A valuable contribution to the research field of sarcopenia. The paper is well written, however, some edits have to be made to improve the clarity and power of the paper. In addition, some issues through ought the paper could be addressed more, see below:
Major comments:
The aim of the study does not seem to match the title of the paper. The use of the term ‘effect’ does not seem to be correct in the context of the study. In my opinion, ‘effect’ can only be used in intervention studies. In this case, influence or association would be more appropriate. Phrasing like in line 141-143 would be more appropriate. In table 1. Please add the number of subjects in each protein group. This is now only stated in the first paragraph of the results and makes the table a bit harder to understand. Also, footnote 1 does not appear in the table itself. Table 2 – footnote 1 does not appear in the table. In addition, it is not clear why age is categorized into below and above 60 years of age. This is not specified in the statistical analyses section. Please clarify. The discussion section can be structured better for more clarity. As the main aim of the study is to see what alcohol consumption does on the association between protein intake and muscle mass, it is better to start the discussion with this section. (now at line 285). Also there is quite a repetition of the introduction in that section (line 285 – 297). This could be reduced or restructured. Please discuss the findings presented in figure 1 In the first paragraph of the discussion, it is stated that significant associations were not observed in heavy drinkers, however, looking at table 2. the heavy drinkers has a p-value of 0.024. could this be a typo? Please elaborate more on how limitations of the study may have effected the outcome and what/how this should be done differently in future studies.
Comments per line:
Line 36 – the used reference is the older definition of sarcopenia by the European Working Group on Sarcopenia in Older people, which still refers to older adults. The updated definition (2018) recognizes that sarcopenia can also occur earlier in life. Respected to the age of included subjects the updated definition seems more appropriate, please update the reference.
Line 55 – declined should be decline
Line 55 – 57 - sentence is unclear, please rephrase
Line 63 – little is known suggests that at least we know something, so I expect to see at least 1 reference. Please add a reference
Line 65- 66 – sentence does not add towards the aim of the study, please remove sentence
Line 67- 69 – The aim of the study does not seem to match the title of the paper. The use of the term ‘effect’ does not seem to be correct in the context of the study. In my opinion, ‘effect’ can only be used in intervention studies. In this case, influence or association would be more appropriate. Phrasing like in line 141-143 would be more appropriate.
Line 106 – please add reference where the categorization is based on.
Line 161 – 163 please rephrase, the ‘in addition’ does not refer to the sentence before
Line 230 – please clarify/rephrase ‘expressed body weight’.
Line 271- 272 – please clarify which associations are being mentioned here. In addition, it is not clear on which numbers this statement is based.
Line 339 – subject should be subjects
Author Response
We thank you for your time and expertise in reviewing the manuscript.
All changes have been made according to the reviewer’s comments in the text. Detailed explanations we have made are described in the rebuttal letter we enclosed.
The English in this manuscript has been checked by at least two professional editors, both native speakers of English.
Thank you for your consideration for our manuscript.
Yours sincerely,

Round 2
Reviewer 2 Report
Dear Author,
Thank you for your work and considering my comments, the manuscript improved significantly.
with kind regards